# Experimentally validated design principles of heteroatom-doped-graphene-supported calcium single-atom materials for non-dissociative chemisorption solid-state hydrogen storage

Yong Gao[1], Zhenglong Li[1], Pan Wang [2], Wen-Gang Cui[1], Xiaowei Wang[3], Yaxiong Yang[1], Fan Gao[1], Mingchang Zhang[1], Jiantuo Gan[1], Chenchen Li[1], Yanxia Liu[1], Xinqiang Wang[1], Fulai Qi[1], Jing Zhang[2], Xiao Han[2], Wubin Du[4], Jian Chen[5] ✉, Zhenhai Xia [6] ✉ & Hongge Pan[1] ✉

Non-dissociative chemisorption solid-state storage of hydrogen molecules in host materials is promising to achieve both high hydrogen capacity and uptake rate, but there is the lack of non-dissociative hydrogen storage theories that can guide the rational design of the materials. Herein, we establish generalized design principle to design such materials via the first-principles calculations, theoretical analysis and focused experimental verifications of a series of heteroatom-doped-graphene-supported Ca single-atom carbon nanomaterials as efficient non-dissociative solid-state hydrogen storage materials. An intrinsic descriptor has been proposed to correlate the inherent properties of dopants with the hydrogen storage capability of the carbon-based host materials. The generalized design principle and the intrinsic descriptor have the predictive ability to screen out the best dual-doped-graphene-supported Ca single-atom hydrogen storage materials. The dual-doped materials have much higher hydrogen storage capability than the sole-doped ones, and exceed the current best carbon-based hydrogen storage materials.

There is an urgent need to transform energy resources from fossil fuels to clean energy to reduce greenhouse gas emissions all over the world[1]. Hydrogen is promising, as an alternative to fossil fuels[2], for future clean energy sources with many inherent advantages such as high heating value (142 MJ/kg), earth-abundance, non-toxicity and eco-friendliness[3]. One of the key issues hindering its widespread application is its storage.

Traditionally, hydrogen is stored in high-pressure gaseous-state and low-temperature liquid-state, but these hydrogen storage technologies (HSTs) are unsafe and costly for storage and transportation.

Solid-state hydrogen storage, as a key link of hydrogen economy, stands out from HSTs by virtue of unique advantages mainly consisting of high gravimetric and volumetric hydrogen storage densities and

[1]Institute of Science and Technology for New Energy Xi'an Technological University, Xi'an 710021, China. [2]School of Materials Science and Engineering, Northwestern Polytechnical University, Xi'an 710072, China. [3]Department of Materials Science and Engineering, University of North Texas, Denton, TX 76203, USA. [4]School of Materials Science and Engineering, Zhejiang University, Hangzhou 310058, PR China. [5]School of Materials Science and Chemical Engineering, Xi'an Technological University, Xi'an 710021, China. [6]Australian Carbon Materials Centre, School of Chemical Engineering, University of New South Wales, Sydney, NSW 2052, Australia. ✉e-mail: chenjian@xatu.edu.cn; zhenhai.xia@unsw.edu.au; Honggepan@zju.edu.cn

safety in practical application such as fuel cell vehicles (FCVs)[4]. However, current solid-state hydrogen storage materials (SHSMs) are still in its infancy[5]. Until now, despite many potential SHSMs available, there are few materials with excellent compressive performances comprising high gravimetric and volumetric capacities, superior cycling stability, fast kinetics, near-ambient operating conditions, high safety and low cost[6–8].

At present, SHSMs can generally be divided into two types: (i) intercalation-type SHSMs such as $MgH_2$, $AlH_3$, $LiBH_4$ etc. based on dissociative chemisorption of $H_2$ molecules, and (ii) physisorption-type SHSMs as a result of non-dissociative physisorption mainly represented by graphene, porous carbon, carbon nanotubes (CNTs) etc[9–15]. The intercalation type has sluggish kinetics, inferior cycling stability and harsh operating conditions, originating from high dissociation energy barrier of $H_2$ molecule on the surface and slower diffusion and stronger chemical bond of hydrogen atoms in bulk intercalation-type SHSMs[16]. Although the physisorption type can provide excellent cycle stability and fast hydrogen adsorption/desorption kinetics, its hydrogen storage density is relatively low due to weak van der Walls adsorption of $H_2$ molecules[17].

One of the promising solutions to the above issues is to use non-dissociative chemisorption solid-state hydrogen storage materials (NC-SHSMs). This is a special form of adsorption that lies between physical adsorption and chemical bond, with partial charge transfer resulting into elongated but unbreakable hydrogen bond of hydrogen molecule. This type of materials has an advantage of the intercalation-type SHSMs to achieve higher hydrogen storage density while having fast kinetics comparable to those of physisorption-type SHSMs. Such advantageous features of NC-SHSMs originate from surface-induced chemisorption with the moderate adsorption energy between physical and chemical adsorption[18–20]. However, until now, except for few intrinsic NC-SHSMs such as amorphous metal hydride and titanium oxide based on Kubas effect[21–24] and MOF and zeolite originating from the interaction between metal sites and $H_2$ molecules[25–28], there is almost no extrinsic NC-SHSMs available for hydrogen storage, although it possesses greater space and freedom for structural design and modification (Supplementary Tables 1 and 2).

Graphene, as a typical 2D carbon nanomaterial with such inherent merits as high thermal and chemical stability, extraordinary strength and toughness, corrosion-resistance and extremely high specific surface area[29], can be modified via the introduction of various dopants[30], the loading of appropriate metal single atoms or nanoparticles[31] and even their combination to achieve extraordinary extrinsic non-dissociative $H_2$ storage contribution[32]. Moreover, calcium, as an earth-abundant alkaline-earth metal element, has been considered as a superior metal single atom anchored on (doped-)graphene for non-dissociative hydrogen storage because of its low cohesive energy (1.8 eV) between bulk Ca and appropriate binding energy between Ca single atoms and $H_2$ molecules[33–36].

However, to our best knowledge, there has been no report on systematic metal-free-heteroatom-doped-graphene-supported Ca single-atom carbon nanomaterials as NC-SHSMs except for a few partially related investigations from perspective of theoretical calculation (Supplementary Table 2). More significantly, so far, in-depth insight into intrinsic physical principle governing the impact of different dopants on non-dissociative hydrogen storage contribution has not well been elucidated, which restricts reclamation of application potential of such carbon-based extrinsic NC-SHSMs (Supplementary Tables 1 and 2). To rationally design high-efficient carbon-based extrinsic NC-SHSMs, it is necessary to develop design principles to correlate the carbon nanostructures with the hydrogen storage properties. Although some work has been done in developing non-dissociative hydrogen storage theories, none of them is able to describe $H_2$ adsorption/desorption mechanism at atomic and electronic levels, and thus cannot predict the properties of materials (Supplementary Table 3).

In this study, we proposed an experimentally validated generalized design principle to quantitively describe non-dissociative hydrogen storage behaviors of heteroatom-doped-graphene-supported Ca single-atom NC-SHSMs. The effect of different dopants on hydrogen storage was systematically studied via the first-principles calculation, structural characterization and performance test for heteroatom-doped-graphene-supported Ca single-atom NC-SHSMs. Finally, a novel descriptor that was verified by the focused experiments was established to predict hydrogen storage properties of single-/dual-doped-graphene-supported Ca single-atom NC-SHSMs including hydrogen storage density and rate, which was in line with experimental results. The proposed generalized design principle and descriptor may open a new window for non-dissociative hydrogen storage field.

## Results

### Optimal sites and adsorption process of $H_2$ storage on X-G-Ca

Three basic graphene models (G), including nanosheet and armchair- and zigzag-nanoribbons (Supplementary Fig. 1), were used to establish all possible 38 heteroatom-doped graphene models (X-G) via the introduction of p-block elements X (X = B, P, Sb, Si, N, O, S, I, Br, Cl and F) (Fig. 1a). Ca single atom was then placed at all possible hollow sites of each X-G model to generate X-G-Ca models (Supplementary Calculation methods and Supplementary Figs. 2–5), and the binding energies $E_b$ of Ca single atom chemisorbed on all possible 200 hollow sites of X-G models were calculated using the DFT methods to evaluate the stability of X-G-Ca models with the most negative binding energy. The calculation results illustrated that the most stable Ca single atom in chemisorption generally appeared around X-dopants or near graphene edges mainly due to doping and edge effects (Fig. 1b).

Hydrogen adsorption/desorption processes were further studied via the DFT calculations. In adsorption process, $H_2$ molecules diffuse to X-G-Ca surfaces and adsorb at the active sites under the external driving conditions such as low temperature (T) and/or high pressure (P). Upon changing the external conditions (e.g., increased T and/or reduced P), the $H_2$ molecules desorb (release) from adsorption sites. The adsorption/desorption processes can be described by Eq. (1).

$$H_2 + * \underset{Desorbing}{\overset{Adsorbing}{\rightleftharpoons}} H_2^* \tag{1}$$

where * represents the adsorption sites of X-G-Ca surface.

The adsorption energy was calculated for the adsorption of $H_2$ molecule on all possible 60 sites of the 10 most stable X-G-Ca models (Supplementary Figs. 2–5). The optimal sites of $H_2$ storage (adsorption) with the most negative adsorption energy change $\Delta E$ (Supplementary Figs. 2–5) or the minimum adsorption Gibbs free energy change $\Delta G_{H2^*}$ (Fig. 1b), were screened out, all of which were located on the top sites of Ca single atoms on X-G-Ca models. As can be seen in Fig. 1b, the introduction of dopants improves the adsorption strength of $H_2$ molecules compared with undoped-graphene-supported Ca single-atom models.

According to the calculation results of the optimal $H_2$ storage (adsorption) sites on X-G-Ca, the hydrogen storage (adsorption) possibly follows such a process that $H_2$ molecules firstly adsorb at the optimal sites with the lowest adsorption energy barrier, and occupy these sites with the second lowest adsorption energy barrier, at last store at the sites with the highest adsorption energy barrier under certain external driving conditions. Thus, we added $H_2$ molecules continuously to Ca site on the 10 most stable X-G-Ca models to simulate the practical $H_2$ storage process, as shown in Fig. 1d. Basically, this adsorption process can be divided into three stages: i) Stage 1, non-dissociative chemosorption (Fig. 1c, 1–5), in which 5 $H_2$ molecules directly adsorb on the Ca atom with a distance of ~2.4 Å between Ca atom and adsorbed $H_2$ molecules; ii) Stage 2, the surface physisorption (Fig. 1c, 6–15), at which $H_2$ molecules occupy these sites away from the

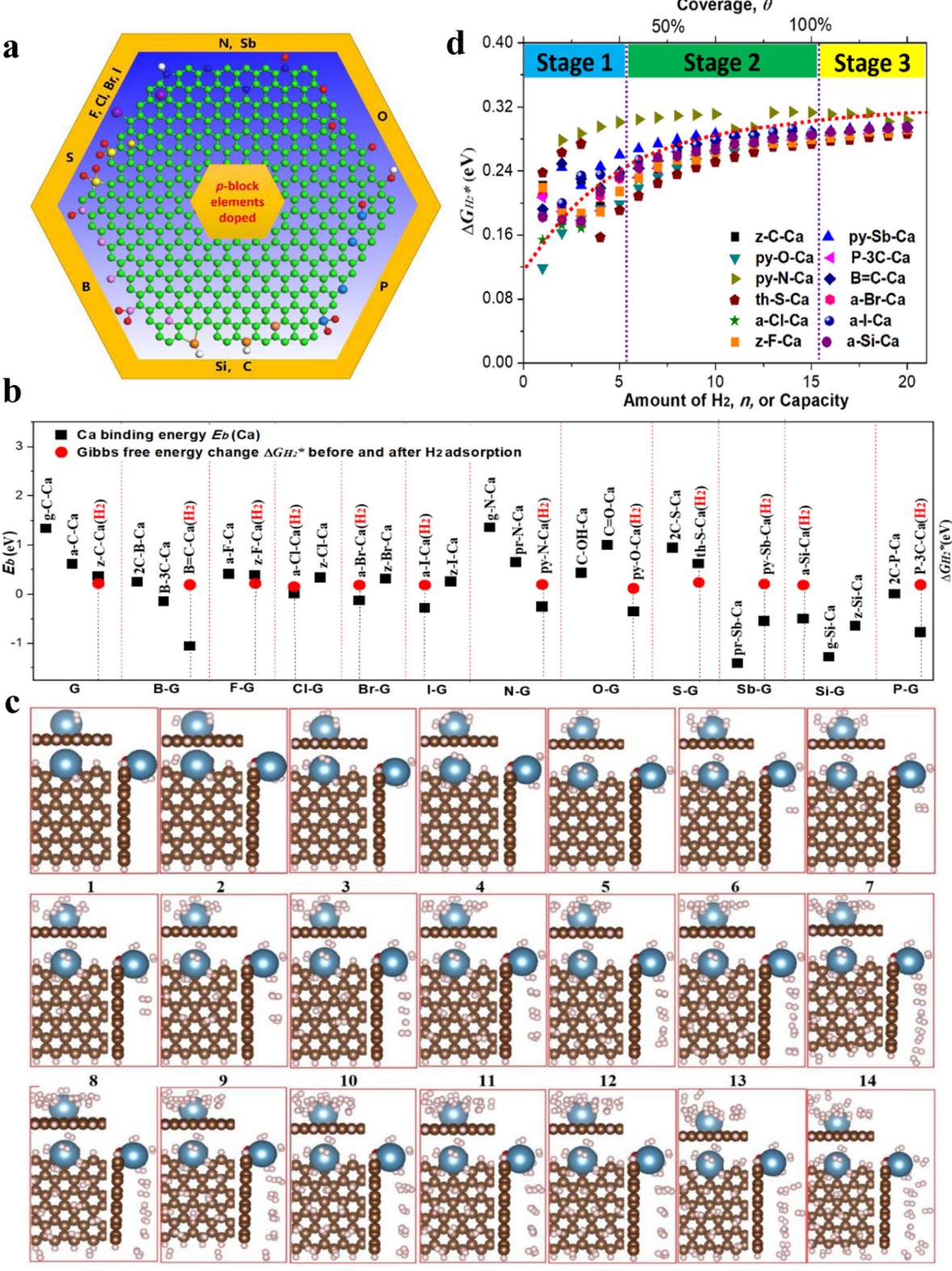

**Fig. 1 | H₂ molecule adsorption calculation and storage mechanism. a** Summary of the heteroatom doping configuration: (clockwise) pr-N(S$_b$), py-N(S$_b$), g-N(S$_b$), N(S$_b$)-O, py-O, C-O-C, C = O, C-OH, P-3C-O, P-3C, py-P, P-2C-2O, g-C(S$_i$), z-C(S$_i$) and a-C(S$_i$), B-3C, C-B-2O, B = C, py-B, B-O, th-S(−2O), py-S(2 O) and C-S-C, z and a-(F, Cl, Br and I) and g-(F, Cl, Br and I). green/gray, blue, pink, light cadet blue, yellow, red, purple, and white balls represent C(S$_i$), N(S$_b$), B, P, S, O, F (Cl, Br and I), and H atoms, respectively. **b** The most negative binding energy $E_b$(Ca) of Ca atom chemisorbed on each X-G substrate and the corresponding minimum adsorption Gibbs free energy change $\Delta G_{H2^*}$ of H₂ molecule adsorbed on each optimal X-G-Ca substrate. **c** Structural evolution upon H₂ molecules adsorbed continuously on O-G-Ca substrate as the best one in all X-G-Ca substrates. **d** The change of H₂ molecules adsorption Gibbs free energy ($\Delta G_{H2^*}$) under standard conditions as the function of the number ($n$) or capacity of H₂ molecules adsorbed on optimal X-G-Ca substrates. Here, red dotted line serves as energy change trend, and stage 1, stage 2 and stage 3 correspond to non-dissociative chemosorption, the surface physisorption and weak physisorption, respectively.

Ca single atom, but near the graphene with the average distance of ~3.4 Å between the $H_2$ and graphene substrate; and iii) Stage 3, weak physisorption (Fig. 1c, 16–20) in which the second layer of $H_2$ molecules is formed upon the full coverage ($\theta = 1$). The adsorption Gibbs free energy change $\Delta G_{H2*}$ at standard conditions of $H_2$ molecules were also calculated as the function of coverage $\theta$ (or the amount of adsorbed $H_2$ $n$ or capacity). As can be seen in Fig. 1d, with increasing the $H_2$ uptake, $\Delta G_{H2*}$ gradually increases and finally levels off at Stage 3. At Stages 1 and 2, the increase of $\Delta G_{H2*}$ can be ascribed to different adsorption strength between $H_2$ molecules and adsorption sites and increasing lateral interaction among $H_2$ molecules at coverage $\theta < 1$, while the nearly constant $\Delta G_{H2*}$ is mainly due to weaker physisorption strength between $H_2$ layers at $\theta > 1$.

### Generalized design principle of $H_2$ storage capacity and rate

Leveraging our previous theoretical model of energy level filling proposed in calcium-ion batteries (CIBs)[37], we put forward the generalized design principle of $H_2$ storage capacity and rate to further guide the rational design of X-G-Ca non-dissociative solid-state $H_2$ storage materials. In short, according to the insight into $H_2$ storage process of X-G-Ca unit cells, we assume that there are different regions with respective energy barriers of $H_2$ adsorption on X-G-Ca surface (Fig. 2a). In the process of $H_2$ storage, $H_2$ molecules driven by external conditions (temperature $T$ and pressure $P$) firstly adsorb on these regions with the lowest energy barrier (Generally, these regions are Ca atoms and these closest regions to Ca atoms), and adsorb in these regions with the second lowest energy barrier until these regions with the highest energy barrier are occupied by $H_2$ molecules, as schematically shown in Fig. 2a.

Moreover, with the increase of adsorption energy barrier of regions, $H_2$ storage process becomes more difficult. Therefore, it is reasonable to assume that the amount of $H_2$ molecules adsorbed on different regions follows the Boltzmann distribution. Under certain conditions ($T$ and $P$), the specific uptake capacity $C_{H_2/site}$ and rate $v_{H_2/site/s}$ of $H_2$ storage on X-G-Ca surface can be derived and given by Eqs. (2) and (3) (See the details in Supplementary Calculation Methods), respectively.

$$
C_{H_2/site} = \begin{cases}
\dfrac{-k_B T \left( e^{-\frac{\Delta G_{H_2^*}^{max}}{k_B T}} - e^{-\frac{\Delta G_{H_2^*}^{min}}{k_B T}} \right)}{\left( \Delta G_{H_2^*}^{max} - \Delta G_{H_2^*}^{min} \right) - k_B T \left( e^{-\frac{\Delta G_{H_2^*}^{max}}{k_B T}} - e^{-\frac{\Delta G_{H_2^*}^{min}}{k_B T}} \right)}, & \Delta G_{H_2^*}^{min} > 0 \\[3em]
\dfrac{k_B T \left( 1 - e^{-\frac{\Delta G_{H_2^*}^{max}}{k_B T}} \right)}{\left( \Delta G_{H_2^*}^{max} - \Delta G_{H_2^*}^{min} \right) - k_B T \left( e^{-\frac{\Delta G_{H_2^*}^{max}}{k_B T}} - e^{-\frac{\Delta G_{H_2^*}^{min}}{k_B T}} \right)}, & \Delta G_{H_2^*}^{min} < 0
\end{cases}
\tag{2}
$$

$$
v_{H_2/site/s} = \begin{cases}
\dfrac{-\lambda k_0 c_{H_2} k_B T \left( e^{\frac{\Delta G_{H_2}^{max}}{k_B T}} - e^{\frac{\Delta G_{H_2}^{min}}{k_B T}} \right)}{\left( \Delta G_{H_2}^{max} - \Delta G_{H_2}^{min} \right) + k_B T \left( e^{\frac{\Delta G_{H_2}^{max}}{k_B T}} - e^{\frac{\Delta G_{H_2}^{min}}{k_B T}} \right)}, & \Delta G_{H_2^*}^{i} > 0 \\[3em]
\dfrac{\lambda k_0 c_{H_2} k_B T \left( 1 - e^{-\frac{\Delta G_{H_2}^{max}}{k_B T}} \right) \left[ \Delta G_{H_2^*}^{max} - k_B T \left( e^{\frac{\Delta G_{H_2}^{max}}{k_B T}} - 1 \right) \right]}{\left[ \left( \Delta G_{H_2^*}^{max} - \Delta G_{H_2^*}^{min} \right) - k_B T \left( e^{-\frac{\Delta G_{H_2}^{max}}{k_B T}} - e^{-\frac{\Delta G_{H_2}^{min}}{k_B T}} \right) \right] \left[ \Delta G_{H_2^*}^{max} + k_B T \left( e^{\frac{\Delta G_{H_2}^{max}}{k_B T}} - 1 \right) \right]}, & \Delta G_{H_2^*}^{i} < 0
\end{cases}
\tag{3}
$$

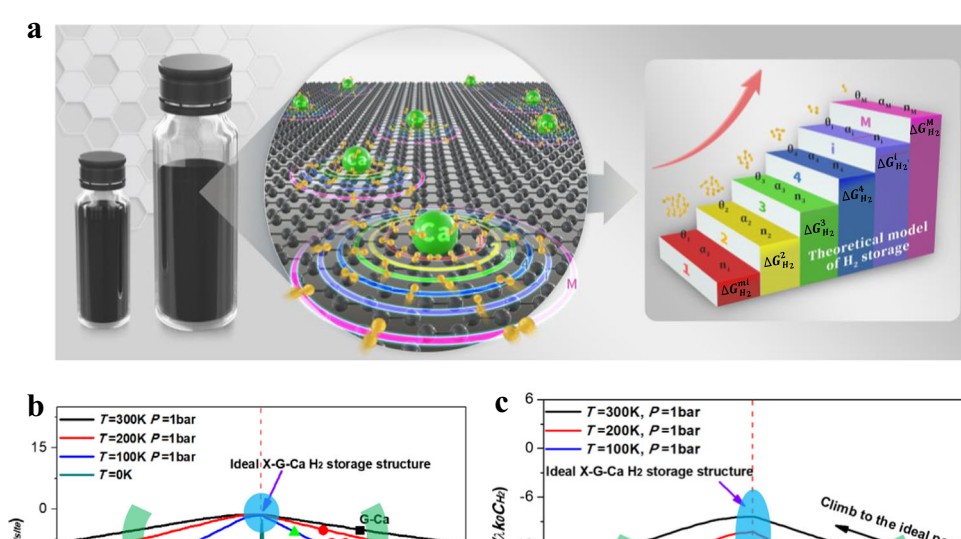

**Fig. 2 | The theoretical model of $H_2$ storage and trend in intrinsic storage capacity and rate of X-G-Ca carbon-based materials. a** The proposed theoretical model of generalized design principle. The volcano trend of $H_2$ storage capacity (**b**) and rate (**c**) versus the minimum Gibbs free energy change under various external conditions consisting of $T = 300$ K, $T = 200$ K, $T = 100$ K and $T = 0$ K at given pressure $P = 1$ bar. Here, black, red and green dots in (**b**) and (**c**) refer to predicted properties for G-Ca models. The blue circular shadows indicated by the purple arrows in (**b**) and (**c**) indicate the ideal X-G-Ca $H_2$ storage structure. The black, red and blue arrows in (**b**) represent the trend of properties with the decrease of the Gibbs free energy change. The thick arrows in (**b**) and (**c**) indicate the trend of properties with the decrease of absolute temperature $T$.

where $\Delta G_{H_2^*}^{\max}$ and $\Delta G_{H_2^*}^{\min}$ are the maximum and minimum values of the Gibbs free energy change of $H_2$ molecule adsorbed on site of each particular X-G-Ca structure surface, respectively, $k_B$ and $T$ are the Boltzmann constant and the absolute temperature, respectively, and $\lambda$ is the pre-factor for Boltzmann distribution, and $k_0$ is the reaction rate constant, and the concentration of $H_2$ molecules is constant for any given $H_2$ storage system.

Equations (2) and (3) not only establish the relationship between $H_2$ storage properties ($C_{H_2/site}$ and $v_{H_2/site/s}$) and intrinsic factors ($\Delta G_{H_2^*}^{\min}$) and extrinsic factors ($T$ and $P$), but also provide a theoretical base or design principle for simultaneously improving $H_2$ storage capacity and rate of X-G-Ca materials. Clearly, both the $H_2$ storage capacity and rate strongly rely on $\Delta G_{H_2^*}^{\min}$, which can be tuned by modifying the X-G-Ca structures, and controlling the external conditions ($T$ and $P$). More specifically, more negative $\Delta G_{H_2^*}^{\min}$ represents stronger interaction between $H_2$ and X-G-Ca surface, which leads to more $H_2$ storage. In this case, the $H_2$ uptake is exothermic without harsh external driving conditions such as lower temperature $T$ and/or higher pressure $P$, but it is not conducive to release of $H_2$ molecules. Whereas the more positive value, i.e., weaker interaction between $H_2$ and X-G-Ca surface, leads to less $H_2$ storage because of endothermic reaction, but it is easier for $H_2$ to release. Therefore, only when the interaction is neither too strong nor too weak, can the $H_2$ adsorption/desorption performance be optimal. Thus, there is a volcano relationship between the storage capacity and $\Delta G_{H_2^*}^{\min}$, as shown in Fig. 2b, c. The stronger interaction is located at the left side of the volcano ($\Delta G_{H_2^*}^{\min} < 0$), where $H_2$ molecules cannot be released, while the weaker interaction is situated at the right side ($\Delta G_{H_2^*}^{\min} > 0$), where $H_2$ molecules can store but require proper external conditions to overcome the energy barrier (e.g., black, red and green dots in Fig. 2b, c). Only when $\Delta G_{H_2^*}^{\min}$ climbs to the summit of the volcano plot ($\Delta G_{H_2^*}^{\min} = 0$), can $H_2$ adsorption/desorption performance of X-G-Ca approach the ideal value (black, red and blue arrows in in Fig. 2b, c). Thus, the generalized design principle provides the direction how to enhance $H_2$ storage capacity and rate of non-dissociative solid-state $H_2$ storage materials beyond current $H_2$ storage theories (Supplementary Table 3).

More significantly, the volcano curves become steeper with the decrease of $T$ at a constant $P$, as can be seen in Fig. 2b, c, illustrating that decreasing $T$ can increase $H_2$ storage capacity by reducing $\Delta G_{H_2^*}^{\min}$, limited at $T = 0$ K. Moreover, $H_2$ adsorption/desorption rate decreases as $T$ lowers, as illustrated downward movement of volcano curves in Fig. 2c, implying that decreasing $T$ will lead to lower $H_2$ adsorption/desorption rate mainly due to inferior low-temperature activity.

## Climbing to volcano peak according to proposed intrinsic descriptor $\Phi$

Although the above volcano-shaped relationship can be used to predict the $H_2$ storage capability of the X-G-Ca NC-SHSMs, it is required to carry out extensive DFT calculation work for screening out the optimal $H_2$ storage materials. Therefore, we have identified an innovative descriptor to correlate the $H_2$ storage capacity with the intrinsic properties of the heteroatoms $X$, given by Eq. (4).

$$\varnothing = \frac{E_X A_X R_X / N_X}{E_C A_C R_C / N_C} \qquad (4)$$

where $E$, $A$, $R$, and $N$ are the electronegativity, electron affinity, radius, the number of outermost electrons, respectively, and the subscripts $X$ and $C$ refer to dopant and carbon, respectively.

As shown in Fig. 3a and Supplementary Table 4, $\Delta G_{H_2^*}^{\min}$ as a function of the descriptor $\Phi$ forms a volcano-shaped relationship, indicating that the proposed descriptor $\Phi$ is capable of capturing the nature of $\Delta G_{H_2^*}^{\min}$. Interestingly, as shown in Fig. 3b and Supplementary Fig. 6, for $H_2$ storage on $p$-block element $X$ ($X$ = B, P, Sb, Si, N, O, S, I, Br, Cl, F) doped-graphene-supported Ca single-atom (X-G-Ca) NC-SHSMs, $H_2$ storage properties, i.e., specific capacity ($C_{H_2/site}$) and rate ($v_{H_2/site/s}$), also exhibit reverse volcano relationship as a function of descriptor. Specifically, the $H_2$ storage properties of the X-G-Ca NC-SHSMs at the summit of the volcano plot are the worst rather than the best, which is different from these descriptors proposed in our previous works about electrode materials and catalyst[38,39]. In other words, for this volcanic relationship, the farther the descriptor $\Phi$ is to the top of the volcano plot, the better the $H_2$ storage properties of such doped-graphene-supported Ca single-atom (X-G-Ca) NC-SHSMs is. Consequently, guided by descriptor $\Phi$, the $H_2$ storage performances on X-G-Ca NC-SHSMs continuously climb towards the submit of the volcano plot resulting from the generalized design principle as proposed above, as shown in Fig. 3c, d. Thus, according to the generalized design principle, oxygen-doped-graphene-supported Ca single-atom (O-G-Ca) NC-SHSMs at the higher position of the volcano plot should exhibit the best $H_2$ storage properties among all X-G-Ca NC-SHSMs based on the predictions from the viewpoint of $\Delta G_{H_2^*}^{\min}$. Conclusively, the intrinsic descriptor $\Phi$, consisting of physical parameters obtained conveniently in elemental periodic table, has the predictive capability for $H_2$ storage properties.

## Origin of volcano plot guided by descriptor $\Phi$

The proposed design principle provides a theoretical base for guiding rational design of the materials to achieve superior $H_2$ storage capacity $C_{H_2/site}$ and rate $v_{H_2/site/s}$ by optimizing $\Delta G_{H_2^*}^{\min}$ toward zero (Fig. 3a). The origin of this volcano relationship can be understood via the insight into the interactions between $H_2$ molecule, Ca single-atom and doped-graphene structures. While $\Delta G_{H_2^*}^{\min}$ represents the interaction strength between $H_2$ molecule and Ca single atom, it also strongly depends on the charge redistribution of doping structures. Therefore, the intrinsic descriptor $\Phi$ is correlated with physical parameters of dopants including radius $R_X$, electronegativity $A_X$, electron affinity $E_X$ and the number of outermost electrons $N_X$ of heteroatoms $X$, which would capture the essence of this complex interaction (or $\Delta G_{H_2^*}^{\min}$). We have calculated the differential charge distribution for each optimal X-G-Ca structure with $H_2$ molecule adsorbed on optimal site (X-G-Ca-$H_2$) and the corresponding bond length change of $H_2$ molecule (Fig. 4a, b, Supplementary Figs. 15–20 and Supplementary Table 6) to establish their correlation with descriptor $\Phi$ (Supplementary Fig. 21). It was found the amount of charge transferred to $H_2$ molecule from Ca atom for each X-G-Ca-$H_2$ model as a function of descriptor, $\Phi$, exhibited the similar volcano relationship as shown in Fig. 3a (Fig. 4c). This implied that $\Delta G_{H_2^*}^{\min}$ strongly relied on the amount of charge transferred to $H_2$ molecule from Ca atom, and this dependency is an inverse relationship as shown in Supplementary Fig. 22d. Moreover, this volcano relationship can be roughly ascribed to the different ability of doped-graphene substrate (X-G) and $H_2$ molecule to compete for the charges of Ca single atom for each X-G-Ca-$H_2$ structure, as shown in Supplementary Fig. 23.

To gain more insights into the origin of descriptor $\Phi$ from perspective of bond orbital, the density of states (DOSs) of X-G, X-G-Ca and X-G-Ca-$H_2$ models were calculated to estimate the adsorption strength (Supplementary Fig. 26 and Supplementary Tables 8–10). As shown in Fig. 4d, both the Ca atom and $H_2$ adsorptions will cause left shift of DOS distribution of the doped-graphene structure (P-3C), but the former has more effect compared to the latter. This is mainly due to stronger chemisorption of Ca on the doped-graphene compared with weaker non-dissociative chemisorption between $H_2$ and Ca. This difference in adsorption strength originates from the difference in

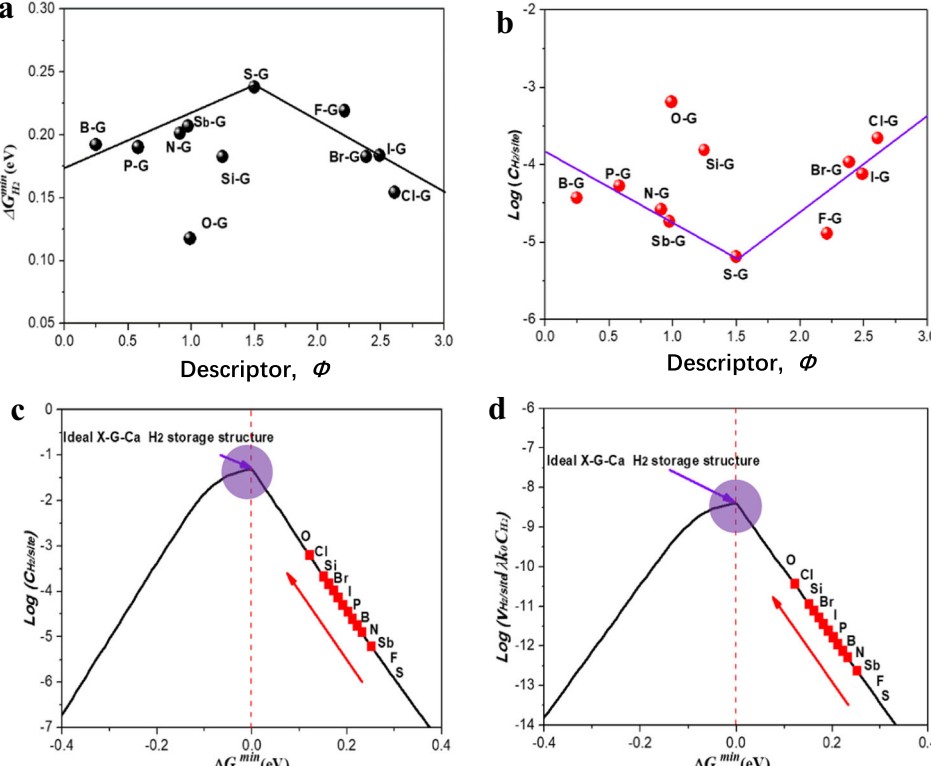

**Fig. 3 | The minimum Gibbs free energy change before and after H₂ adsorbed on the optimal site of each X-G-Ca model and normalized H₂ storage capacity of each optimal X-G-Ca structure versus descriptor Φ. a** The minimum Gibbs free energy change before and after H₂ adsorbed on the optimal site of each X-G-Ca model versus descriptor. **b** the predicted H₂ storage capacity against the descriptor. Climbing to the submits of volcano curves (namely parameter related to (**c**). H₂ storage capacity ($log\ (C_{H_2/site})$) and (**d**) rate ($log(v_{H_2/site/s}\ /\ \lambda k_O C_{H2})$) curves by optimizing the minimum Gibbs free energy change $\Delta G_{H_2}^{min}$ at standard conditions ($T = 300\ K$ and $P = 1\ bar$) based on intrinsic descriptor Φ. Here, black and red dots in (**a**) and (**b**) represent different X-G-Ca models. Red dots in (**c**) and (**d**) indicate the H₂ storage properties of different X-G-Ca models and red arrows refer to the trend of H₂ storage properties of different X-G-Ca models. The purple circular shadows indicated by the purple arrows in (**c**) and (**d**) indicate the ideal X-G-Ca H₂ storage structure.

electronic structures. Additionally, three different DOSs of Ca, X-G, X-G-Ca and X-G-Ca-H₂ structures were determined to find the strongest correlation between DOSs and descriptor Φ (Supplementary Figs. 28–29 and Supplementary Tables 7–10). The results show that DOS2 of Ca atom chemisorbed on X-G substrates can be extracted to form a well-ordered volcano-shaped relationship (Fig. 4e) mainly due to the fact that Ca single atom acts as the bridge of interaction between H₂ and X-G.

The above relationship can well be elucidated by molecular orbital theory. As shown in green zone in Fig. 4f, firstly, for X-G-Ca structures, the new hybridized electron orbitals as a result of the interaction between Ca with electron orbital $\sigma_1$ and chemisorption site with electron orbital $\sigma_2$ are spilt into bond orbital $\sigma$ and anti-bond orbital $\sigma^*$, and the anti-bond orbital with higher energy level $\sigma^*$ is generally in an incompletely filled state, which can continue to interact with H₂ molecule with electron orbital $\sigma_3$, finally splitting into new bond orbital $\sigma_N$ and anti-bond orbital $\sigma_N^*$, as highlighted by blue zone in Fig. 4f. Whereas, the adsorption strength of H₂ is determined by the filling degree of new anti-bond orbital $\sigma_N^*$, which depends on the level of anti-bond orbital energy. Empty or excess of anti-bond orbital will lead to an inappropriate adsorption strength, which is either too strong or too weak. Thus, DOS2 controls the level of anti-bond orbital energy, and thus determine bond strength, which can serve as the essence of descriptor Φ. More importantly, the filling degree of new anti-bond orbital $\sigma_N^*$ can be tuned via the modification of electron orbital of doped-graphene support $\sigma_2$, as shown in Fig. 4f. This is why the introduction of dopants into graphene can improve the adsorption strength of H₂ molecules on X-G-Ca NC-SHSMs.

## Synthesis and characterizations of focused doped-graphene-supported Ca single-atom NC-SHSMs

To validate the proposed generalized design principle, three heteroatom-doped-graphene-supported Ca single-atom samples (X-G-Ca, X = N, O, P) were synthesized. The synthetic method of atomically dispersed Ca supported on nitrogen-doped graphene (N-G-Ca) is schematically illustrated in Fig. 5a. First, N atoms were introduced into graphene by adopting melamine ($C_3H_6N_6$) as the nitrogen source (see Materials synthesis in Methods). After that, calcium nitrate ($Ca(NO_3)_2$) was fully mixed with nitrogen-doped graphene (N-G) in 600 ml of deionized water. After stirring for 12 h, the mixture was centrifuged to extract the black slurry. Finally, the black slurry was dried in vacuum freezing drying oven, yielding the nitrogen-doped-graphene-supported Ca single-atom samples. Additionally, oxygen-doped graphene (O-G) was obtained via high-temperature reduction of graphene oxide (GO) in tube furnace, and phosphorus-doped graphene (P-G) was fabricated via the same method as synthesis of N-G. Then, dispersed Ca single atoms anchored on O-G and P-G substrates were prepared according to the same process (Fig. 5a).

The structures and morphologies of the materials were examined using scanning electron microscopy (SEM). As shown in Fig. 5b and Supplementary Fig. 7, similar morphologies were observed in all the doped-graphene-supported Ca single-atom samples. This relatively consistent morphology was further confirmed via transmission electron microscopy (TEM) images (Fig. 5c and Supplementary Fig. 8), demonstrating that morphology of the doped-graphene substrate was well preserved without noticeable morphological change or destruction.

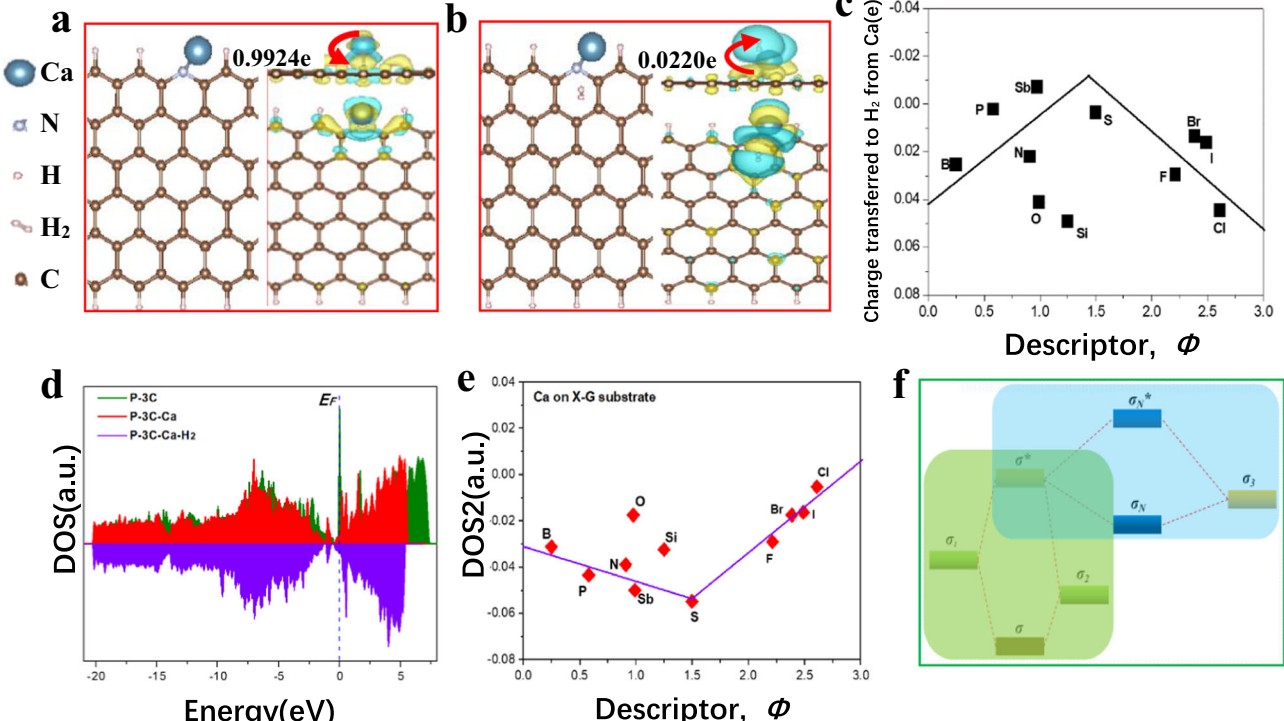

**Fig. 4 | Charge transfer and electronic structure of H₂ stored on doped-graphene-supported Ca single atoms (X-G-Ca) surface.** Atomic structure and corresponding differential charge density and Bader charge transfer of (**a**) the optimal N-doped-graphene-supported Ca single atom (N-G-Ca) and (**b**) H₂ adsorbed on the optimal site of N-G-Ca. **c** Bader charge transferred from Ca to H₂ for each optimal X-G-Ca(·H₂) as the function of intrinsic descriptor $\Phi$. Blue color indicates positive charge and yellow color indicates negative values of electrons quantities. The isosurface value is set to 0.003 e/Bohr³. Two red arrows indicates the direction of charge transfer. **d** DOS change among X-G, X-G-Ca and X-G-Ca(-H₂) structural models. **e** the volcano relationship between DOS2 and descriptor $\Phi$. Here, DOS and DOS2 are the density of state in overall range of energy and the weighted DOS center in overall range of energy, respectively. $\Phi$ is intrinsic descriptor and is given by Eq. (4). **f** Energy level diagram that illustrates two orbital hybridization involving the first hybridization of electronic states between doped-graphene(X-G) and Ca single atom (purple area) and the second hybridization of electronic states between optimal H₂ storage sites and Ca chemisorbed on X-G surface (green area). $E_F$ refers to Fermi level, and σ and σ* indicate bonding and anti-bonding states, respectively.

Moreover, the energy dispersive spectroscopy (EDS) elemental mapping of the samples directly showed the homogeneous distribution of C, N and Ca elements and other heteroatoms elements (Fig. 5d and Supplementary Figs. 8 and 9). Additionally, the atomic structures of the materials were also studied with the aberration-corrected high-angle annular dark-field scanning transmission electron microscopy (AC-HAADF-STEM). As shown in Fig. 5e, f, atomically dispersed Ca single atoms (yellow circles) were uniformly anchored on the N-G support. Subsequently, no large nanoparticles were observed in the high-resolution TEM image of N-G-Ca (Fig. 5g).

X-ray photoelectron spectroscopy (XPS) was adopted to unveil the chemical composition and structure of each element in the samples. For N-G-Ca sample, XPS survey spectrum firstly revealed the coexistence of C, N and Ca elements coming from graphene matrix, heteroatoms precursors and single-atom Ca sources (Fig. 6a and Supplementary Fig. 11), which confirmed the successful introduction of heteroatoms and Ca single atoms. Moreover, from the XPS spectra, the atomic percentage of the dopants (N, O, P) in the samples was obtained, ranging from 2% to 13%, which was in line with the EDS results (Supplementary Table 5). The X-G bond types were also analyzed with high resolution XPS. As shown in Fig. 6b, N-G graphene has three nitrogen species consisting of central graphitic nitrogen (g-N) and edge pyridinic (py-N) and pyrrolic N (pr-N) in the graphene plane. The bond types of other doped-graphene substrates were also investigated (Supplementary Fig. 11). In addition to heteroatom, the Ca 2p XPS spectra were also obtained for N-G-Ca (Fig. 6c), and the Ca 2p₁/₂ and Ca 2p₂/₃ peaks located at 352.8 and 347.8 eV, respectively, but metallic Ca 2p₃/₂ spectrum (344.9 eV) cannot be found. Thus, Ca exists

in the form of single atoms not nanoparticles in nitrogen-doped-graphene[40]. More information on Ca single atom supported on different doped-graphene was gained by X-ray diffraction (XRD) and N₂ adsorption-desorption isotherms. The XRD results of different Ca-adsorbed samples were almost the same as that of pure carbon black, disclosing no corresponding diffraction peak of Ca nanoparticles (Fig. 6d), which are consistent with AC-HAADF-STEM images and XPS results, indicating the successful preparation of various doped-graphene-supported Ca single-atom samples. In addition, the nitrogen adsorption-desorption isotherm of different samples demonstrated that the specific surface areas of N-G-Ca, O-G-Ca, and P-G-Ca were 325, 395 and 58 m²/g, respectively (Fig. 6e and Supplementary Fig. 12). The larger surface areas provided more storage sites of H₂ molecules and contributed to the adsorption and desorption of H₂ molecules.

**Experimentally validated design principle and descriptor $\Phi$**

To validate the design principles derived from fundamental considerations, H₂ adsorption/desorption properties of these as-prepared N-G-Ca, O-G-Ca and P-G-Ca samples were evaluated (Fig. 7a and Supplementary Fig. 13). The results show that the samples can store and release H₂ at 77 K by controlling pressure (Fig. 7a). Moreover, reservable adsorption/desorption rate curve (Fig. 7b) suggested that H₂ storage of the samples are surface-induced adsorption and there are almost no structural changes on the surface of these samples in the process of H₂ adsorption/desorption. Cyclic performance indicates a small performance degradation after 6 cycles (Fig. 7c and Supplementary Fig. 13), illustrating this material has superior stability during

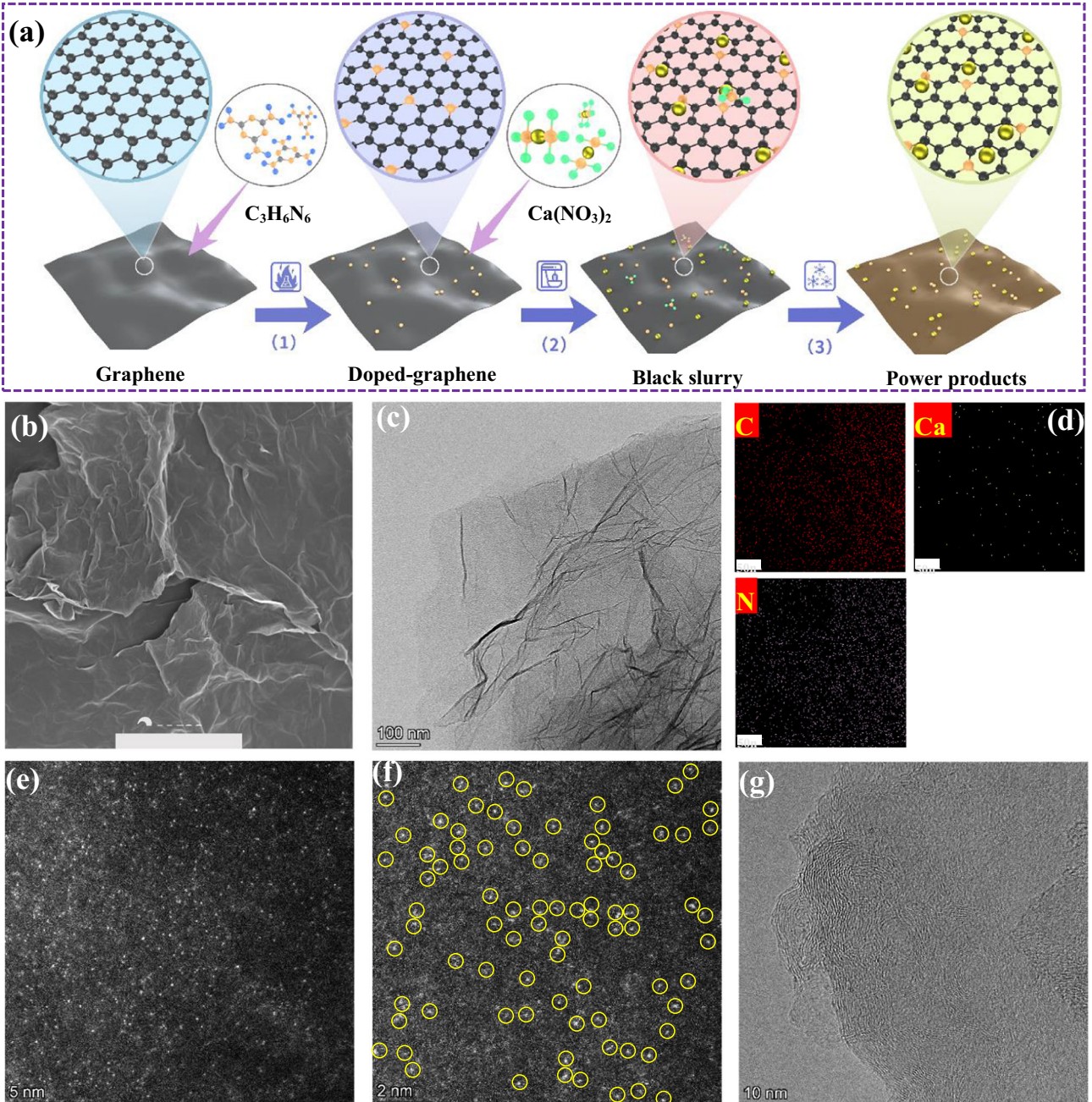

**Fig. 5 | Synthesis strategy and electron microscopic characterization.**
**a** Synthesis route of Ca atomically dispersed on N-doped graphene substrate. (1) The introduction of N atoms. (2) The introduction of atomically dispersed Ca single atoms. (3) Freeze-dry step. **b** SEM image. **c**, **d** TEM image and corresponding EDS elemental mapping of N-doped-graphene-supported Ca single atoms (N-G-Ca). **e**, **f** Aberration corrected HAADF-STEM images of N-doped-graphene-supported Ca single atoms (N-G-Ca) in different scale bars. Here, yellow circles represent the labeled Ca single atoms anchored on N-G-Ca substrate. **g** HR-TEM image of N-doped-graphene-supported Ca single atoms (N-G-Ca).

$H_2$ storage/release. Thus, considering their similar microstructure under the same characterization conditions, it is reasonable to assume that the observed differences in specific capacity for the X-G-Ca samples originates only from dopant concentration (at%) and Ca loading content (at%).

To compare with the theoretical predictions, the $H_2$ storage specific capacity per $H_2$ storage site, $C_{O/site}{}^*$ ($H_2$ site$^{-1}$) was further calculated based on the measured hydrogen content $C_m$ (Supplementary Note 1, Supplementary Table 5 and Supplementary Fig. 13). The trend between calculated results $ln(C_{O/site}{}^*)$ and $\Delta G_{min}$ was shown in Fig. 7c, which showed the similar trend with the volcano relationship from the proposed generalized design principle (Fig. 3c). Moreover, the order of

the calculated $C_{O/site}{}^*$ for N-G-Ca, O-G-Ca and P-G-Ca also followed the similar volcano trend governed by descriptor $\Phi$ (Fig. 3c). Conclusively, both the proposed design principle and intrinsic descriptor were demonstrated by normalized $H_2$ storage property tendency of as-prepared carbon-based Ca single-atom samples.

**Design principles of dual-doped-graphene-supported Ca single-atom NC-SHSMs**

According to the above exploration, it is obvious that the $H_2$ storage capability of the heteroatom-doped-graphene-supported Ca single-atom host material is governed by intrinsic factors (e.g., $\Delta G_{H_2^*}^{min}$) or extrinsic factors (e.g., single-atom Ca loading content, surface area,

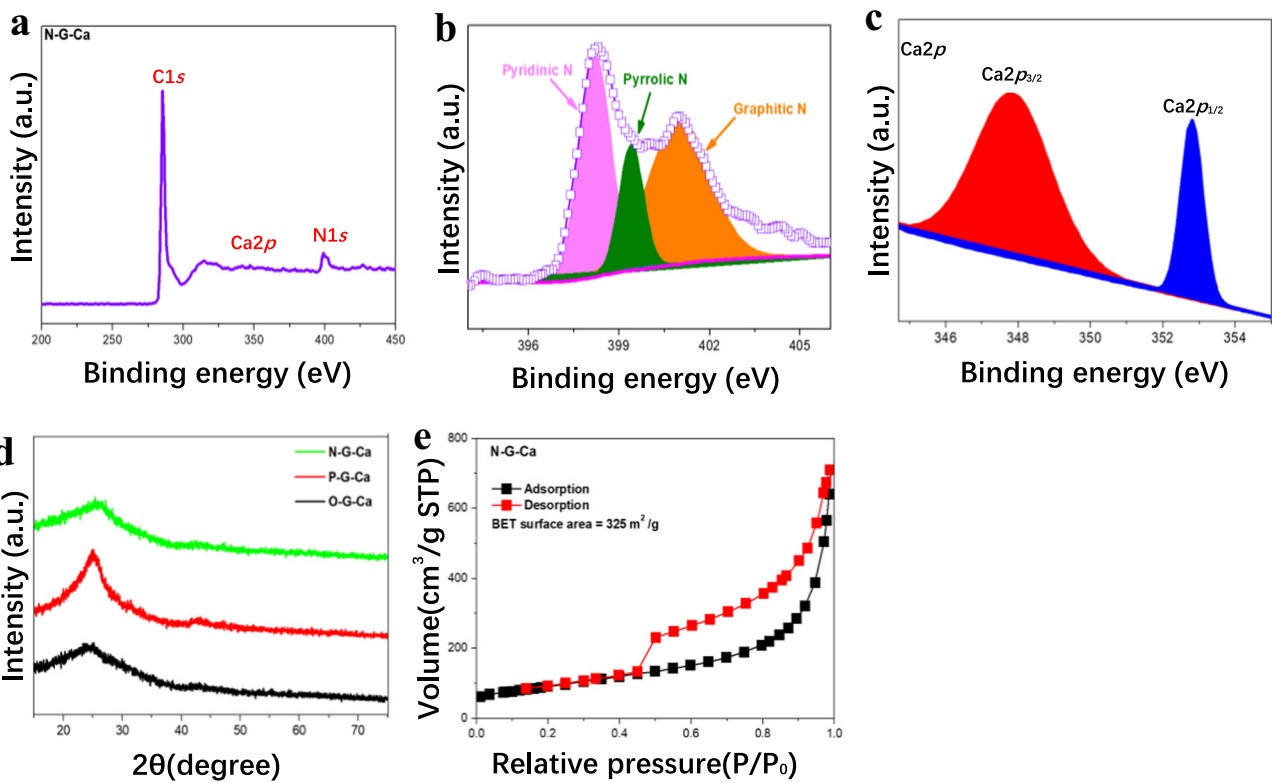

Fig. 6 | **Spectral characterization. a** XPS Survey spectra of N-G-Ca sample. **b**, **c** High-resolution XPS spectra of N-G-Ca and Ca $2p_{1/2}$ and Ca $2p_{2/3}$, respectively. **d** XRD patterns of samples including O-G-Ca, P-G-Ca, N-G-Ca and S-N-G-Ca. **e** Nitrogen adsorption isotherms and BET surface area for N-G-Ca samples.

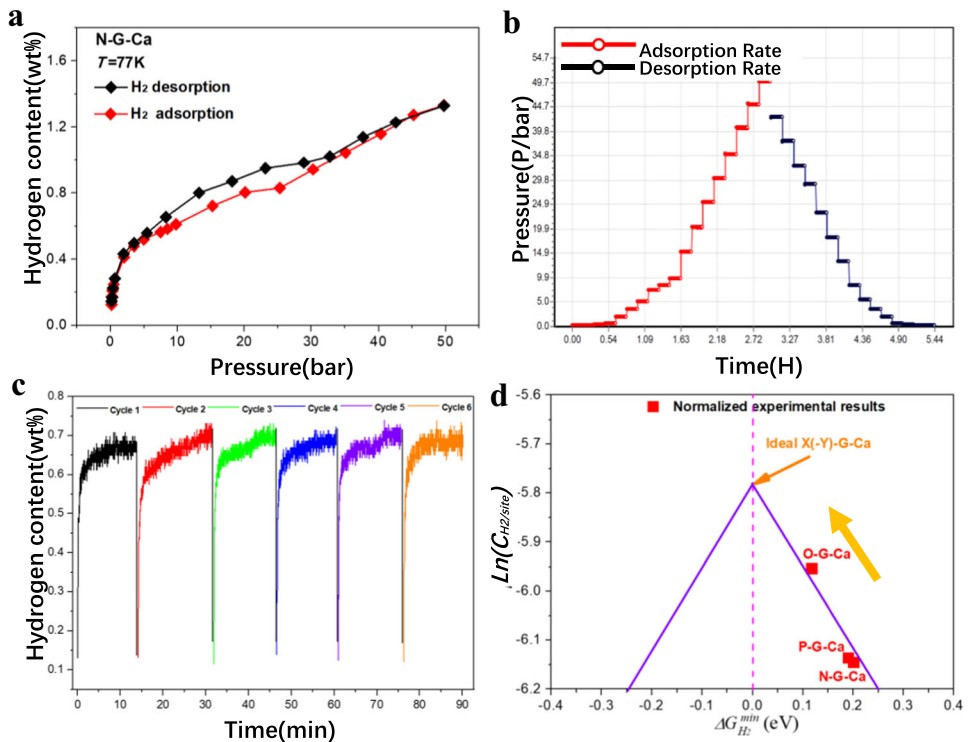

Fig. 7 | **Mutually verified DFT calculation and H₂ storage performance measurements. a** H₂ adsorption/desorption curve as function of pressure. **b** H₂ adsorption/desorption rate. **c** Cyclic performance test of H₂ adsorption/desorption for N-G-Ca sample under $T = 77$ K and $P = 1.2$ MPa. **d** Design principle validated by normalized H₂ storage capacity (Supplementary Table 5). Here, the thick yellow arrow in (**d**) represents the trend of H₂ storage properties, and the thin arrow in (**d**) refer to the ideal X(-Y)-G-Ca model with the best H₂ storage properties.

and heteroatom doping concentration). Thus, we propose a two-step strategy to further enhance $H_2$ molecules storage ability on the NC-SHSMs. The first step is to optimize $\Delta G_{H_2^*}^{min}$ closer to zero by multi-doping strategy. Since S element at the top of volcano plot is the critical one among all the heteroatoms used as dopants in this study (Fig. 3b), it was selected to combine with other $p$-block elements to form dual-doped-graphene-supported Ca single-atom (X-Y-G-Ca) models (Supplementary Fig. 14).

We have constructed 154 possible models for each dual-doped graphene configuration, and screened out 9 optimal X-Y-G-Ca models from all possible 790 single atom Ca adsorption sites via DFT calculations. The optimal $H_2$ molecule adsorption sites were determined from all possible 47 $H_2$ adsorption sites on each optimal X-Y-G-Ca models. Finally, the minimum Gibbs free energy change $\Delta G_{H_2^*}^{min}$ was calculated for $H_2$ molecule adsorbed on the optimal dual-doped-graphene-supported Ca single-atom structure (Supplementary Figs. 16–19 and Supplementary Table 4). It was found that $\Delta G_{H_2^*}^{min}$ correlates with the descriptor $\Phi$, in the form of an invert-volcano relationship (Fig. 8a) compared to that of the single-doped-graphene-supported Ca single-atom models (Fig. 3a), which is consistent with the predictions for catalytic activity and $Ca^{2+}$ ions storage capability of dual-doped graphene for both fuel cells and Ca-ion batteries[37,41]. On contrast, for the X-Y-G-Ca, the smaller the difference in descriptor between the two dopants, the better the $H_2$ storage performance of the dual-doped-graphene-supported single-atom Ca (red dotted arrow in Fig. 8a), which is mainly ascribed toparticular volcano relationship, namely dopant S acting as the critical point (the worst doping type) located at the top of this volcano plot for single-doped-graphene-supported Ca single-atom models. Consequently, as shown in Fig. 8b, c, the specific capacity and rate of $H_2$ storage as a function of descriptor $\Phi$ led to two volcano-shaped relationships that could be used to guide the design of dual-doped-graphene-supported Ca single-atom NC-SHSMs. According to the relationship and $\Delta G_{H_2^*}^{min}$, it was predicted that $H_2$ molecules storage ability for S-Si-dual-doped-graphene-supported Ca single-atom structure (S-Si-G-Ca) ($\Delta G_{H_2^*}^{min} = 0.084$ eV) has surpassed that of all sole-doped-graphene-supported Ca single-atom (X-G-Ca) NC-SHSMs (Supplementary Table 4). The capacity per unit $H_2$ storage site $C_{O/site}$ for S-Si-G-Ca increases by four-fold compared with the best single-doped-graphene-supported Ca single-atom (e.g., O-G-Ca) NC-SHSMs.

The origin of the invert-volcano plot for X-S-G-Ca NC-SHSMs can be understood from the synergistic effect of dual-doped-graphene-supported Ca single-atom substrates. As long as the optimal adsorption strength between Ca and dual-doped graphene is achieved via controlling the appropriate combination of two dopants according to the guidance from dual-doped volcano plot, can the NC-SHSMs reach the best $H_2$ storage properties. The synergistic effect can be further elucidated from viewpoint of microscopic electro-magnetic force and molecular orbital theory. As mentioned above, the interaction between $H_2$ and Ca single atom chemisorbed on doped-graphene strongly depends on charge distribution of Ca single atom, as shown comparison of charge transfer before and after $H_2$ molecule adsorption in Fig. 8g, h. The regular relationship can be captured. As shown in Fig. 8d, e, there is the volcano relationship between the charge transfer from Ca to $H_2$ and the descriptor $\Phi$ as well as that between DOS2 (the weighted DOS center of each optimal X-Y-G-Ca(-$H_2$) below Fermi level) and the descriptor $\Phi$. These relationships are consistent with the $H_2$ storage property-descriptor relationship, suggesting that the interactions between them are involved in the synergistic effect.

The another strategy to improve the $H_2$ storage capability is to increase single-atom Ca loading content, heteroatom doping concentration and surface area of $H_2$ host materials. According to the established expressions for the hydrogen storage capacity $C_{H_2/site}$ and rate $v_{H_2/site/s}$ (intrinsic factor), under certain conditions ($T$ and $P$), the total hydrogen storage capacity ($C_{H_2/M-sites}$) and rate ($v_{H_2/M-sites/s}$) considering the extrinsic factors (total sites of X-G-Ca surface) can roughly be determined from

$$C_{H_2/M-sites} = \mu C_{H_2/site} \cdot \frac{S_a \cdot m_a}{2S_c} \cdot \alpha \qquad (5)$$

$$v_{H_2/M-sites/s} = \mu v_{H_2/site/s} \cdot \frac{S_a \cdot m_a}{2S_c} \cdot \alpha \qquad (6)$$

where $S_a$ is the surface area per unit mass hydrogen storage sample materials (in unit of $m^2 g^{-1}$), $m_a$ is the mass of sample materials (unit, g), $S_c$ is set to be the area per carbon atom on graphene (around $2.6 \times 10^{-20}$ $m^2$) due to the fact that smaller amount of calcium single atoms chemisorbed on doped graphene, the pre-factor 2 is due to the fact that $H_2$ molecule storage site can be simply regarded as hollow site of doped-graphene-supported Ca single-atom surface, implying that a $H_2$ molecule occupies averagely 2 net carbon atoms on doped-graphene-supported Ca single-atom surface, $\alpha$ is the load atomic percentage of calcium single atoms decorated with X-G surface, and $\mu$ is the dimensionless quantity serving as a correction factor to tune the error between experimental results and theoretical prediction for any particular $H_2$ storage system. Here, it deserves to note that although doping concentration has an impact on specific capacity of $H_2$ storage, its role is realized mainly via tuning metal Ca single atoms. Therefore, compared with Ca single atom loading, contribution of heteroatom doping concentration to specific capacity can be roughly ignored. Thus, as long as single Ca atoms do not aggregate, both the $H_2$ storage capacity and rate can be enhanced by increasing surface area $S_a$, Ca loading content $\alpha$ and even doping concentration for a given graphene-based $H_2$ storage host materials.

With these two strategies, the dual-doped-graphene-supported Ca single-atom NC-SHSMs can exhibit a capacity comparable to and even beyond that of the best carbon-based $H_2$ storage materials currently available. As shown in Fig. 8f., the as-prepared N-G-Ca, O-G-Ca and P-G-Ca graphene-based NC-SHSMs has already outperformed the current carbon-based $H_2$ storage materials (See normalized process in Supplementary Note 2). Moreover, it is predicted that the $H_2$ storage properties of dual-doped S-Si-G-Ca NC-SHSMs far exceed all the carbon-based materials (See prediction process in Supplementary Note 3) based on the calculated $\Delta G_{H_2^*}^{min}$ (-0.084 eV) under the same physical properties including BET surface area (325 $m^2 g^{-1}$), Ca loading mass (0.33 at%), doping concentration (7.61 at%), and external conditions (77 K and 2 MPa), same as that of as-prepared N-G-Ca NC-SHSMs in our experiment.

Notably, the superior $H_2$ storage properties can be achieved via dual-doping in NC-SHSMs (e.g., S-Si-G-Ca), and these materials can be prepared via the same preparation method as the above single-doped-graphene-supported single-atom Ca NC-SHSMs. Importantly, the designed dual-doped-graphene-supported single-atom Ca NC-SHSMs has also an unmatched $H_2$ storage rate and cycle ability beyond all other intercalation-type $H_2$ storage materials due to the nature of surface-induced non-dissociative chemisorption.

## Discussion

In summary, we have proposed the design principles that govern non-dissociative $H_2$ storage in carbon materials based on the theoretical calculation and experimental verification. A descriptor $\Phi$, which closely correlates with inherent properties of dopants in graphene materials, was defined, from which the guiding principles were developed for rational design of doped-graphene-supported Ca materials for $H_2$ storage. The theory can accurately predict non-dissociative $H_2$ storage properties of the materials, which was confirmed by the focused experiment. Based on the guidance of descriptor $\Phi$, volcano and invert-volcano relationships for sole- and

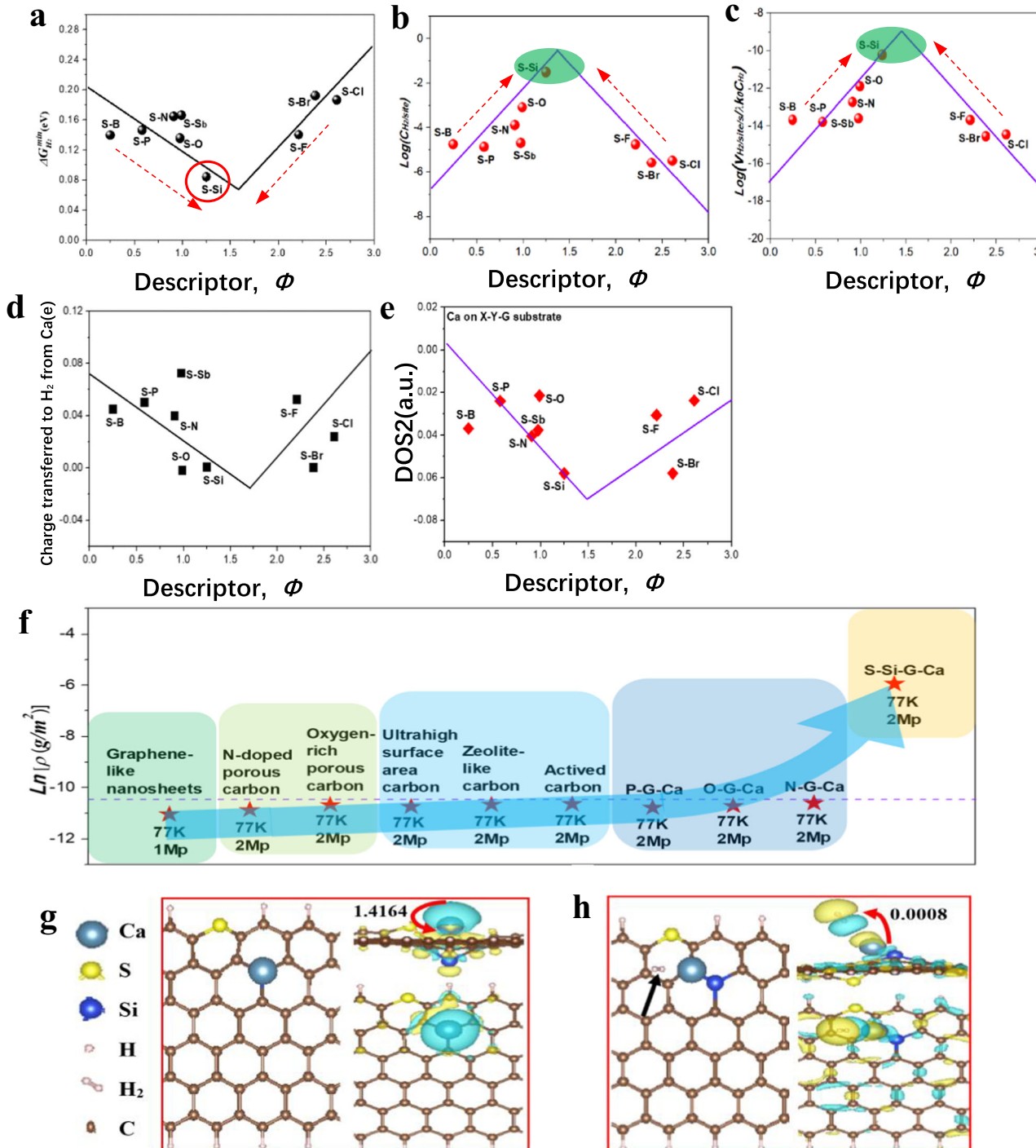

**Fig. 8 | Invert-volcano relationship govern by intrinsic descriptor $\Phi$ and differential charge density distribution and Bader charge transfer on optimal dual-doped-graphene-supported Ca single atoms (X-Y-G-Ca). a** The invert-volcano relationship related to the minimum Gibbs free energy change $\Delta G_{H_2}^{min}$ before and after $H_2$ adsorption and descriptor $\Phi$ for X-Y-G-Ca. **b, c** The predicted $H_2$ storage capacity and rate against the descriptor $\Phi$ for X-Y-G-Ca. Here, the red dashed arrows in (**a**)–(**c**) indicate the trend of $H_2$ storage properties for X-Y-G-Ca models. And the red circle in (**a**) and the shadows in (**b**) and (**c**) the optimal X-Y-G-Ca model with the best $H_2$ storage properties. **d** Bader charge transferred from Ca to $H_2$ for each optimal X-Y-G-Ca(-$H_2$) as the function of intrinsic descriptor $\Phi$. **e** The volcano relationship between DOS2 and descriptor $\Phi$ for each optimal X-Y-G-Ca(-$H_2$). **f** Comparison between predicted and

experimental $H_2$ storage capacity of doped-graphene-supported Ca single atoms in this work and current other $H_2$ storage carbon materials in references (Supplementary Notes 2 and 3). Here, the arrow shows the trend of normalized $H_2$ storage properties improvement. **g** Atomic structure and charge distribution of the optimal S-Si-codoped-graphene-supported Ca single atoms (S-Si-G-Ca). **h** Atomic structure and charge distribution of the optimal S-Si-G-Ca with $H_2$ molecule (Black arrow) adsorbed on optimal site (S-Si-G-Ca-$H_2$). For the corresponding differential charge density and Bader charge transfer in (**g**) and (**h**)., blue color indicates positive charge and yellow color indicates negative values of electrons quantities. The isosurface value is set to 0.0002 e/Bohr$^3$. Here, red arrows in (**g**) and (**h**). indicate the direction of charge transfer and the black arrow in (**h**) refer to the $H_2$ molecule.

dual-doped graphene-supported Ca single-atom NC-SHSMs, respectively, were established, from which the two-step design principle was proposed to further promote non-dissociative chemisorption $H_2$ storage ability. Based on the strategy, the designed materials can match the current best $H_2$ storage materials. Finally, the electronic structures of the materials and complex electron interactions between Ca, dopant and $H_2$ are analyzed, which provides the base and origin for the predictive ability of the descriptor and related design principles.

## Methods
### Materials synthesis
Graphene doped with $X$ ($X$ = N, O and P) samples were synthesized via modified method in ref. 42. Specifically, the precursor with dopant $X$ and single-layer graphene (purchased from Nanjing XFNANO Materials Tech. Co., Ltd. diameter: 0.5–5 μm, thickness: 0.8 nm, single layer ratio: 80%, purity: 99 wt%, specific surface area 500–1000 $m^2\,g^{-1}$) were mixed in the mass ratio of 1 to 16; And, the mixture was heated to 900 °C (825 °C for Sb-doped sample) at heating rate of 5 °C $min^{-1}$ to further keep 3 h to obtain resulting sole-doped graphene sample, where the process was always under in flowing $N_2$ atmosphere. Melamine ($C_3H_6N_6$) and triphenylphosphine (($C_6H_5$)$_3$P) were used as the sources of N and P elements, respectively. All precursors were purchased from Sigma-Aldrich. After that, calcium nitrate (Ca($NO_3$)$_2$) was fully mixed with heteroatom-doped graphene (X-G) in 600 ml of deionized water. After stirring for 12 h, the mixture was centrifuged to extract the black slurry. Finally, the black slurry was dried in vacuum freezing drying oven, yielding the heteroatom-doped-graphene-supported Ca single-atom samples.

### Structural characterization and hydrogen storage performance measurements
The X-ray photoelectron spectra (XPS) were measured on Thermo Scientific Escalab 250Xi XPS equipment. The incident radiation was monochromatic Al Kα X-rays (1486.6 eV). Nitrogen-sorption isotherms were collected on micromeritics apparatus (ASAP 2020 Plus 2.00) at 77 K and specific surface area was calculated via Brunauer-Emmett-Teller (BET) equation. Scanning electron microscopy (SEM) imaging was conducted on a Thermo Scientific Helios 5 CX. Energy dispersive spectroscopy (EDS) was obtained at the same time as SEM measurement. Transmission electron microscopy (TEM) as well as high-resolution transmission electron microscopy (HRTEM) were conducted on Talos F200S G2. Hydrogen storage performances were tested by use of automatic high pressure gas sorption instrument.

### Computational parameters
The screening of optimal active sites, change of adsorption energy $\Delta E$, charge distribution and density of state (DOS) for $H_2$ storage on the surfaces of various doped-graphene supported Ca single-atom models were calculated using DFT in VASP package. The Perdew-Burke-Ernzerhof (PBE) functional within the generalized gradient approximation (GGA) was used to model the electronic exchange correlation energy[38]. The projector augmented wave (PAW) method was used to describe the ionic cores. The cutoff energy was set to be 400 eV for the plane-wave expansion after testing a series of different cutoff energies. The K-points were set to be 4 × 4 × 1 and 4 × 1 × 1 for the graphene nanosheet and nanoribbons, respectively. Denser K-points of 12 × 12 × 1, 12 × 1 × 1 and 1 × 12 × 1 for sheet, zigzag and armchair nanoribbon models were set to calculate the DOS serving as characterization of electronic structures[37].

### Computational models
Based on different configurations of heteroatom-doped graphene demonstrated experimentally and theoretically (Fig. 1a), pure graphene sheet, armchair and zigzag nanoribbon models were constructed to function as substrates doped with dopants (Supplementary Fig. 1). The graphene sheet model was periodic on both the $x$ and $y$ directions consisting of 48 carbon atoms with an infinite large supercell sheet model, and zigzag and armchair nanoribbon models consisting of 48 and 36 carbons, respectively were periodic only on the $x$ direction, and H atoms were added to saturate carbon dangle bonds on $y$ direction[37]. All dual-doped graphene models (X-Y-G) were constructed via all possible combination configurations of two sole-doped configurations. (Supplementary Fig. 14). After optimization for all doped-graphene models, single atom Ca was added to all possible hollow sites on doped-graphene models to search for the most stable X-G-Ca models acting as non-dissociative hydrogen storage materials. For each dual-doped graphene model configuration, all possible single atom Ca chemisorption sites were tested by DFT calculation to search for the most stable X-Y-G-Ca models serving as non-dissociative hydrogen storage materials. (Supplementary Fig. 14).

## Data availability
The data that support the findings of this study are included in the published article (and its Supplementary Information). Source data are provided with this manuscript and is available from the corresponding author upon request. Source data are provided with this paper.

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

## Acknowledgements

Y.G., Z.L.L., W.G.C., Y.X.Y., F.G. and J.T.G. thank the support from the National Key Research and Development Program of China (No. 2021YFB4000601). Y.G. thanks the support from the National Natural Science Foundation of China (No. 22209128).

## Author contributions

Y.G., Z.H.X. and H.G.P. conceived the project. Y.G. performed the DFT computations, hydrogen storage performance measurements, structural characterization and preparation of doped-graphene supported Ca single-atom. P.W. synthesized the doped-graphene. Z.L.L., W.G.C., Y.X.Y., F.G., M.C.Z., J.T.G., Y.X.L., X.Q.W., F.L.Q. and W.B.D. helped analyze the experiment data. J.Z., X.H., C.C.L. and X.W.W. helped with theoretical calculations and analyses. Y.G., Z.H.X. and H.G.P. co-wrote the manuscript. J.C. modified the manuscript. All authors discussed and analyzed the data. All authors have approved the final version of the manuscript.

## Competing interests

The authors declare no competing interests.
