## [Peer Review File · Nature Communications]

Experimentally validated design principles of heteroatom-doped-graphene-supported calcium single-atom materials for non-dissociative chemisorption solid-state hydrogen storageEditorial Note: Figure R1 in this Peer Review File has been amended to remove third-party material where no permission to publish could be obtained.

REVIEWER COMMENTS

Reviewer #1 (Remarks to the Author):

Comments: The authors investigated a series of Ca atoms embedded graphene serving as non-dissociative chemisorption hydrogen storage materials. They propose a novel descriptor Φ that has been experimentally validated and shown superiority compared with previous studies. The revealed principle and descriptor can guide the rational design of solid-state hydrogen storage materials. I would like to recommend it for publication after the following minor revisions.

1. In this study, only the binding energy E_b between Ca single atom and graphene substrate is used to evaluate the stability of the supported Ca single-atoms. Is this enough?
2. Figure 1b is inappropriately placed after 1c and 1d in the discussion. In addition, the three stages of Figure 1b are differentiated by color, and the background colors affect the clarity of the data symbols. It is recommended to use dashed lines to separate the different stages.
3. Regarding the experimental part, it is recommended that the authors provide cycling data of the sample. In addition, the mapping image of Ca is not clear possibly due to the low Ca content on graphene. To demonstrate the uniform distribution of single-atom Ca, the authors could map a region on multilayer graphene.

Reviewer #2 (Remarks to the Author):

This research article presents a comprehensive study on generalized design principles, hydrogen storage mechanisms and origins at atomic and electronic level of heteroatom-doped-graphene-supported Ca single-atom materials for non-dissociative chemisorption solid-state hydrogen storage. This is an interesting, timely and important research topic to solid-state hydrogen storage field, and can arouse a series of follow up works. In particular, the proposed generalized design principles beyond current adsorption-type solid-state hydrogen storage theories can provide a theoretical base for all adsorption-type solid-state hydrogen storage materials. Moreover, the construction of descriptor able to guide rational design of sole- and dual-doped-graphene-supported Ca single-atom materials for non-dissociative solid-state hydrogen storage materials can provide a new idea for the design of single-atom hydrogen storage materials in the future. The research results are universal, solid and convincing. I would be happy to recommend the publication of this meaningful article on Nature Communications after the several minor revisions.

1. These EDS mapping images with poor contrast in Figure 5d cannot provide any information. please modify them.
2. Is the generalized design principle in the main text the same as generalized Sabatier principle in the Supplementary Table 3? If so, please express it uniformly.
3. In this paper, non-dissociative chemisorption of H₂ molecules was investigate. However, there is no clear explanation or definition for non-dissociative chemisorption of H₂ molecules.

4. The article used a high-throughput selection method to search for the optimal models, but did not provide the final result of the screening.
5. How do you get the line of volcano? Is it a linear regression method?
6. To provide readers with more information on Ca functionalized carbon for hydrogen storage, the following paper can be cited:
Theoretical Study of Hydrogen Storage in Ca-Coated Fullerenes, *J. Chem. Theory Comput.* 2009, 5, 374–379.

Response to the reviewers' comments

We thank the reviewers for their time and effort in reviewing our manuscript and for providing valuable suggestions for improvements. We also follow the reviewer's suggestions and revised the paper.

Reviewer #1: The authors investigated a series of Ca atoms embedded graphene serving as non-dissociative chemisorption hydrogen storage materials. They propose a novel descriptor Φ that has been experimentally validated and shown superiority compared with previous studies. The revealed principle and descriptor can guide the rational design of solid-state hydrogen storage materials. I would like to recommend it for publication after the following minor revisions.

1. In this study, only the binding energy E_b between Ca single atom and graphene substrate is used to evaluate the stability of the supported Ca single-atoms. Is this enough?

R: Thanks for your question. Indeed, there are more criteria for determining the stability of a material structure from the perspective of theoretical calculation, such as cohesive energy, formation energy, etc. However, for this type of heteroatom-doped-graphene-supported Ca single atom structures, measuring its stability through the binding energy E_b is one of the most reliable methods due to the fact that the stability of this material is mainly determined by the bonding strength between the metal single atom and the substrate.

2. Figure 1b is inappropriately placed after 1c and 1d in the discussion. In addition, the three stages of Figure 1b are differentiated by color, and the background colors affect the clarity of the data symbols. It is recommended to use dashed lines to separate the different stages.

R: Thanks for your suggestion. Following your suggestions, we have corrected the order of Figure 1 and redrawn the Figure 1b by removing the background color (See the highlight on pages 4-6)

3. Regarding the experimental part, it is recommended that the authors provide cycling data of the sample. In addition, the mapping image of Ca is not clear possibly due to the low Ca content on graphene. To demonstrate the uniform distribution of single-atom Ca, the authors could map a region on multilayer graphene.

R: Thanks for your suggestion. According to your suggestions and in conjunction with relevant articles about cyclic testing (10 cycles around, Progress in Materials Science 88 (2017) 1–48; Adv. Mater. 2020, 2002647; Adv. Mater. 2019, 1902757), we have provided the cycling data of

representative samples (N-G-Ca) and conducted the corresponding data description and analysis to further improve the paper (See Figure 7c and the highlight on page 16 in the main text and Supplementary Fig.13). Basically, cyclic performance indicates a small performance degradation after 6 cycles (Fig. 7c and Supplementary Fig. 13), illustrating this material has superior stability during H₂ storage/release mainly due to non-dissociative chemisorption mechanism.

In addition, we have revised the Figure 5d with a clearer image of Ca distribution according to your suggestions. (See the Figure 5d on page 14).

Reviewer #2: This research article presents a comprehensive study on generalized design principles, hydrogen storage mechanisms and origins at atomic and electronic level of heteroatom-doped-graphene-supported Ca single-atom materials for non-dissociative chemisorption solid-state hydrogen storage. This is an interesting, timely and important research topic to solid-state hydrogen storage field, and can arouse a series of follow up works. In particular, the proposed generalized design principles beyond current adsorption-type solid-state hydrogen storage theories can provide a theoretical base for all adsorption-type solid-state hydrogen storage materials. Moreover, the construction of descriptor able to guide rational design of sole- and dual-doped-graphene-supported Ca single-atom materials for non-dissociative solid-state hydrogen storage materials can provide a new idea for the design of single-atom hydrogen storage materials in the future. The research results are universal, solid and convincing. I would be happy to recommend the publication of this meaningful article on Nature Communications after the several minor revisions.

1. These EDS mapping images with poor contrast in Figure 5d cannot provide any information. please modify them.

R: Thanks for your suggestion. We have revised the Figure 5d to have a clearer EDS image. (See the Figure 5d on page 14).

2. Is the generalized design principle in the main text the same as generalized Sabatier principle in the Supplementary Table 3? If so, please express it uniformly.

R: Following your suggestions, we have checked and corrected the expression of the name of this design principle to make it more unified and standardized (See the highlight in the Supplementary Table 3). In Table 3, the “generalized Sabatier principle” has been changed to “generalized design principle.”

3. In this paper, non-dissociative chemisorption of H₂ molecules was investigated. However, there is no clear explanation or definition for non-dissociative chemisorption of H₂ molecules.

R: Non-dissociative chemisorption is a special form of adsorption that lies between physical adsorption and chemical bond, with partial charge transfer resulting into elongated but the hydrogen bond of hydrogen molecule is not broken. As shown in Figure R1 (*Eur. J. Inorg. Chem.* 2016, 3371-3375), from the perspective of charge transfer and activation of H₂, Kubas mechanism should belong to generalized non-dissociative chemisorption despite different intrinsic charge transfer mechanism. As for our designed materials in this study, they also belong to non-dissociative chemisorption between physisorption and metal hydrides, as illustrated in the following figure. This type of adsorption has an advantage of the intercalation-type materials to achieve higher hydrogen storage density while having fast kinetics comparable to those of physisorption-type materials. We have added more description about the non-dissociative chemisorption of H₂ molecules to further elucidate this special adsorption form in the paper (See the highlight on page 2).

[Redacted]

Figure R1 Types of bonds between the substrate and hydrogen molecules (*Eur. J. Inorg. Chem.* 2016, 3371-3375). Kubas-Enhanced adsorption should belong to non-dissociative chemisorption

4. The article used a high-throughput selection method to search for the optimal models, but did not provide the final result of the screening.

R: The screening results including the optimal doping structure models and the corresponding Gibbs free energy change have been displayed in Figure 1b and Supplementary Table 4 for single-doped graphene models. For the case of dual-doped graphene models, the screening results consisting of the optimal doping structure models and the corresponding the Gibbs free energy change were showed in Supplementary Fig. 14 and 16 and Supplementary Table 4.

5. How do you get the line of volcano? Is it a linear regression method?

R: Thanks for your question. There are two types of volcano curves. One is the volcanoes, which is produced based on derived theoretical relationship (e.g., Fig. 2b, c. Fig.3 c, d). The second one is the volcano curves made against the descriptor Φ as shown in Figure 3a and b and Figure 4c and e. In the second type, the figures usually contain too scattered data, in which linear regression approach is not used, and the lines of the volcano are plotted to capture the trend of the curve change. The line in the second type of volcano curve serves as the role of change trend guidance.

6. To provide readers with more information on Ca functionalized carbon for hydrogen storage, the following paper can be cited: Theoretical Study of Hydrogen Storage in Ca-Coated Fullerenes, J. Chem. Theory Comput. 2009, 5, 374-379.

R: Thanks for your suggestion. We have cited and added the above literature (See the highlight (references 36) on page 25 in our paper) to further improve the paper.

REVIEWERS' COMMENTS

Reviewer #1 (Remarks to the Author):

the authors have well addressed the raised questions. I recommend acceptance.

Reviewer #2 (Remarks to the Author):

The authors made significant improvements in the revised version and answered my the main concerns satisfactorily. I would like to recommend the publication as is.

Response to the reviewers' comments

We thank the reviewers for their time and effort in reviewing our manuscript and for providing valuable suggestions for improvements.

Reviewer #1 (Remarks to the Author):

the authors have well addressed the raised questions. I recommend acceptance.

R. We appreciate the reviewer's positive comment.

Reviewer #2 (Remarks to the Author):

The authors made significant improvements in the revised version and answered my the main concerns satisfactorily. I would like to recommend the publication as is.

R. Thanks for the reviewer's supportive comment.